# Effect of the Sintering Temperature on the Compressive Strengths of Reticulated Porous Zirconia

**Chae-Young Lee** [1,2], **Sujin Lee** [1], **Jang-Hoon Ha** [1,*,†], **Jongman Lee** [1], **In-Hyuck Song** [1]
**and Kyoung-Seok Moon** [2,*,†]

1  Ceramics Materials Division, Korea Institute of Materials Science, 797 Changwondaero, Seongsan-gu, Changwon 51508, Korea; cy916@kims.re.kr (C.-Y.L.); adsj1503@kims.re.kr (S.L.); jmlee@kims.re.kr (J.L.); sih1654@kims.re.kr (I.-H.S.)
2  School of Materials Science and Engineering, Gyeongsang National University, Jinju 52828, Korea
*  Correspondence: hjhoon@kims.re.kr (J.-H.H.); ksky.moon@gnu.ac.kr (K.-S.M.)
†  These authors contributed equally to this work.

**Featured Application: Ceramic Membranes.**

**Abstract:** Porous ceramics have separation/collection (open pore) and heat-shielding/sound-absorbing (closed pore) characteristics not found in conventional dense ceramics, increasing their industrial importance along with dense ceramics. Reticulated porous ceramics, a type of porous ceramic material, are characterized by a three-dimensional network structure having high porosity and permeability. Although there have been numerous studies of porous zirconia, which is already widely used, there are insufficient reports on reticulated porous zirconia, and it is still challenging to improve the compressive strength of reticulated porous ceramics thus far, especially considering that too few studies have been published on this topic. Therefore, we prepared reticulated porous zirconia specimens using the replica template method. In this study, the compressive strength outcomes of reticulated porous zirconia were analyzed by controlling the PPI value (25, 45, 60, and 80 PPI) of the sacrificial polymer template, the average zirconia particle size (as-received, coarse, intermediate, and fine), and the sintering temperature (1400, 1500, and 1600 °C). Consequently, we confirm that it is possible to prepare reticulated porous zirconia with a wide range of strengths (0.16~1.26 MPa) as needed with an average particle size and while properly controlling the sintering temperature.

**Keywords:** reticulated porous zirconia; compressive strength; sintering temperature





## 1. Introduction

Porous materials are a generic term for materials with a pore volume of 15 to 95%. Porous materials have both the intrinsic properties of porous materials, namely, separation/collection (open pore structure) and heat-shielding/sound-absorbing (closed pore structure) capabilities. Accordingly, their industrial importance is increasing. Among the various types of porous materials, porous ceramics have the inherent properties of porous materials and simultaneously the advantages of ceramics (high thermal resistance and good chemical stability). These properties make it difficult to replace porous ceramics with alternatives, such as porous polymers and metals. For this reason, these materials can be used in a variety of applications.

There are several porous ceramic preparation methods, including the partial sintering method [1], direct foaming method [2,3], sacrificial template method [4,5], and replica template method [6,7]. Among them, the replica template method is a method that can maintain the shape of the sacrificial polymer template (commercial polyurethane foam) used in the preparation process. This is a method in which a polymer template is impregnated with a ceramic slurry, after which the polymer template is removed through a heat-treatment process [8,9]. The porous ceramics prepared by this process can be identified

through their three-dimensional network structure with high porosity (generally more than 90%) and good permeability. Reticulated porous ceramics are restricted by the pore size and the PPI (pores per inch) of the sacrificial polymer template. The pore density of the sacrificial polymer template is expressed as PPI (pores per inch). The higher the PPI, the greater the number of pores and the smaller the pore size. Reticulated porous ceramics have potential applications as diesel particulate filters (DPF), high-temperature insulation materials, molten metal filters, catalyst carriers, and radar-absorbing materials [10–13].

Porous zirconia has been applied to refractory materials, oxygen sensors, and high-temperature structural materials due to its high melting point and excellent wear resistance [14,15]. Porous zirconia also has been used to create new applications, not only through cost reduction but also through lightweight and permeability improvements [16]. Due to the addition of the stabilizers $Y_2O_3$, CaO, MgO, and $CeO_2$ oxides to zirconia, tetragonal and cubic zirconia that exists in a metastable state at room temperature can be produced [17]. Among the stabilized zirconia, yttria-stabilized zirconia has superior properties, such as strength and fracture toughness characteristics, compared to stabilized zirconia with other oxides added [18]. Among them, yttria-stabilized zirconia containing 3 to 5 mol% Yttria exhibit excellent properties and have the advantage of being easy to manufacture. The most thermally stable yttria-stabilized zirconia with high toughness is zirconia stabilized with 3 mol% yttria [19].

Porous zirconia, a type of porous ceramic material, is used in various applications, and a significant amount of experimental data have been collected. However, research on reticulated porous zirconia remains insufficient. Zirconia is generally sintered at 1350 to 1600 °C [20]. Above 1170 °C, a phase transformation occurs according to the temperature of the zirconia. In this study, yttria-stabilized zirconia (YSZ) with the addition of yttria was used. YSZ is well known to prevent the generation of cracks by a phase transformation (tetragonal phase to monoclinic phase) during cooling. Due to these characteristics, the sintering temperature is limited to 1600 °C. In this study, reticulated porous zirconia specimens were prepared using the replica template method. Systematic experiments were conducted to increase the compressive strength of the reticulated porous zirconia by controlling certain processing conditions, in this case the average particle size, viscosity of the zirconia slurry, sintering temperature, and pore density of the sacrificial polymer template. The effects of the processing conditions on the pore characteristics and the overall compressive strength of the reticulated porous zirconia were also investigated in detail.

## 2. Materials and Methods

In this study, commercial polyurethane foam (SKB Tech, Seoul, Korea) samples 20 mm × 20 mm × 20 mm in size with pore densities of 25, 45, 60, and 80 PPI (pores per inch) were used as sacrificial polymer templates. Zirconia slurries were prepared to coat the 25, 45, 60, and 80 PPI sacrificial polymer templates.

Zirconia slurries were prepared with a combination of a dispersant (Dolapix CE 64), thickener (Methyl cellulose) and an organic binder (Polyvinyl alcohol). The dispersant is vital to remove agglomerated zirconia particles and to prepare a uniform zirconia slurry. In previous reports, two types of dispersants were used: Dolapix CE 64 and DARVAN C-N (Vanderbilt Minerals, Norwalk, CT, USA). Dolapix CE 64 and DARVAN C-N are widely used dispersants for dispersing silicon nitride particles, alumina, and zirconia [21–24]. Dolapix CE 64 is a sodium salt of a polycarboxylic acid, and DARVAN C-N is an ammonium salt of polymethacrylic acid, resulting in different mechanical properties. The dispersant selected was Dolapix CE 64 because the results when using Dolapix CE 64 were superior in previous reports.

The samples consisted of 3 mol% yttria-stabilized zirconia (Qingdao Terio Corporation, Qingdao, China) at 67.74 wt.%, with 1.00 wt.% Dolapix CE 64 (Zschimmer & Schwarz GmbH Co., Burgstädt, Germany) as a dispersant, 1.67 wt.% methyl cellulose (Sigma-Aldrich, Darmstadt, Germany) as a thickener, 100 mL distilled water, and with 4.76 wt.% polyvinyl alcohol (molecular weight (Mn) approx. 500, Junsei Chemical, Japan) used as

an organic binder. The mixed zirconia slurries were ball-milled using zirconia balls for 0, 4, 8, and 24 h. The reticulated porous zirconia specimens were prepared via the replica template method. The sacrificial polymer templates were initially soaked in a zirconia slurry until their internal pores became saturated with zirconia particles. Subsequently, the impregnated sacrificial polymer templates were squeezed to remove any excess zirconia slurry, after which the three sample sets were dried in an oven and sintered at 1400, 1500, and 1600 °C for three hours. The viscosities of the zirconia slurry were measured using a rotary viscometer (ViscoQC 300, Anton Paar GmbH, Graz, Austria), and the compressive strength levels of the reticulated porous zirconia specimens after processing at a size of 20 mm × 20 mm × 20 mm were measured with a tensile tester (RB302 Microload, R&B, Daejeon, Korea). The average particle size distributions of the zirconia were determined by a particle size analyzer (LSTM 13 320 MW, Beckman Coulter, Brea, CA, USA). The microstructures were characterized by means of scanning electron microscopy (SEM, JSM-5800, JEOL, Tokyo, Japan). The voltage was set to 20, the working distance set to 9, and the image mode was SE. The pores were characterized by means of mercury porosimetry (Autopore IV 9510, Micromeritics, Norcross, GA, USA).

## 3. Results

In this study, reticulated porous zirconia specimens of 25, 45, 60, and 80 PPI (pores per inch) were prepared using the replica template method. Figure 1a shows the polymer templates of the sacrificial polymer template (commercial polyurethane foam) specimens with pore densities of 25, 45, and 60 PPI. The as-prepared reticulated porous zirconia specimens with pore densities of 25, 45, and 60 PPI, prepared using the replica method, are shown in Figure 1b.

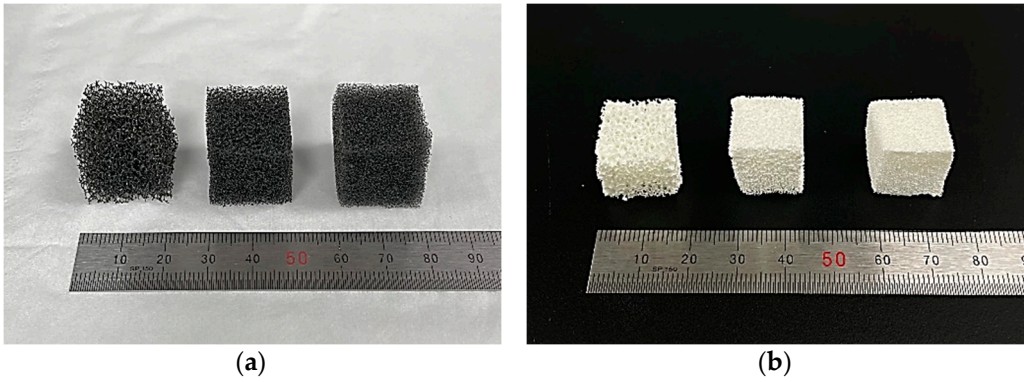

(a)         (b)

**Figure 1.** Optical images of (**a**) the sacrificial polymer templates (commercial polyurethane foam), with pore densities of 25, 45, and 60 PPI, and (**b**) the reticulated porous zirconia specimens, with pore densities of 25, 45, and 60 PPI, prepared through the replica method.

Typical SEM images of the fractured strut walls of the sacrificial polymer template and the reticulated porous zirconia are shown Figure 2a,b, respectively. The triangular voids obtained after the heat treatment and sintering of the sacrificial polymer template, shown in Figure 2b, strongly affect the overall mechanical properties of the reticulated porous ceramics. It is also well known that these inevitable triangular voids lower the compressive strength of reticulated porous ceramics, significantly hindering their application potential. Figure 2b shows the microstructure by which the reticulated porous zirconia with a uniform thickness of the strut wall is formed.

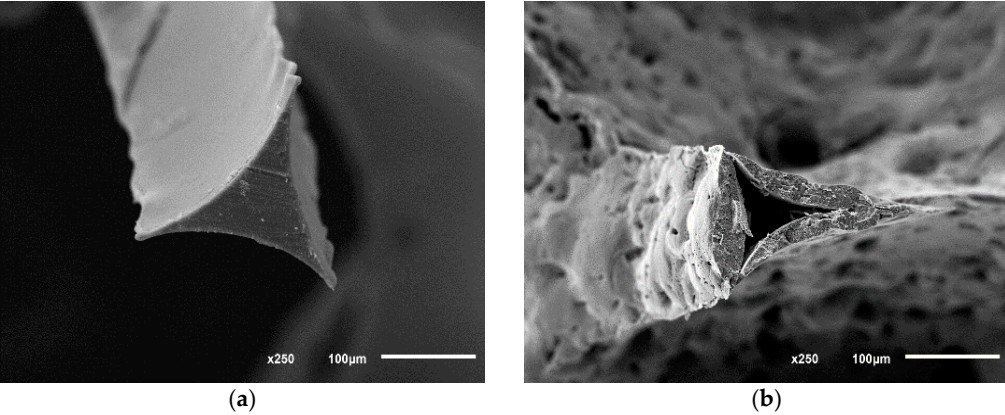

**Figure 2.** Typical SEM images of a fractured strut wall of (**a**) the sacrificial polymer template, and (**b**) a reticulated porous zirconia specimen prepared from fine particles.

The viscosities of the zirconia slurry used for the preparation of the reticulated porous zirconia and the compressive strengths with solid loading rates of 60–67.74 wt.% and with 0.95−1.90 wt.% MC are shown in Figure 3a,b, respectively. This figure shows part of the preliminary experiment, and because it is impossible to proceed with the experiment while changing all of the variables from the beginning, these values function as the basis of how the variables were measured, with the solid loading and MC value used here as variables. The sample with the highest strength was selected as the most optimized slurry (based on 45 PPI). Considering the amount of solid loading as a variable, the compressive strength at a solid loading rate of 67.74 wt.% was the highest. However, this sample did not have enough viscosity to tightly coat the strut walls of the sacrificial polymer template. To control the viscosity only by solid loading, zirconia particles must be added indefinitely until the desired viscosity is reached, though this approach is not efficient. Therefore, after fixing the solid loading amount (67.74 wt.%), the viscosity was controlled by the MC. The MC acts as a thickener that increases in uniformity when mixed with water [25]. It can be seen that the viscosity increased considerably when MC was added. Sintering was performed at 1600 °C to compare the compressive strength of the reticulated porous alumina to that in a previous experiment.

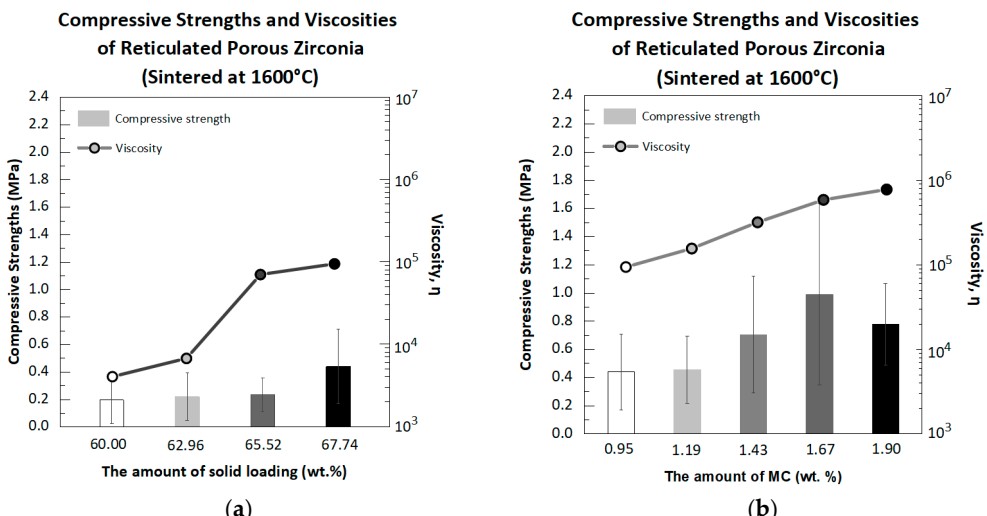

**Figure 3.** Viscosity at a shear rate of 0.01 s-1 of the zirconia slurry, and the compressive strengths: (**a**) 60–67.74 wt.% solid loading; and (**b**) 0.95–1.90 wt.% MC.

In this study, preliminary experiments were conducted based on reticulated porous zirconia with a pore density of 45 PPI because it is difficult to prepare reticulated porous

ceramics with a pore density that exceeds 45 PPI with a sacrificial polymer template. As the pore density of the reticulated porous ceramics approaches 80 PPI, the increase in pore density makes it difficult to coat the internal strut walls of the sacrificial polymer template completely as well. Voids can be generated inside the sacrificial polymer template, reducing the compressive strength.

It is uncertain as to whether the slurry was evenly distributed deep inside the reticulated porous zirconia specimens. Therefore, in this study, a non-destructive μ-CT analysis was utilized to investigate the prepared reticulated porous zirconia specimens. The three-dimensional microstructure of a typical reticulated porous zirconia specimen (pore density = 60 PPI), reconstructed by μ-CT, is shown in Figure 4a. A representative cross-section from the micro-CT reconstructions of the specimen is shown in Figure 4b. This figure indicates that the reticulated porous zirconia specimen had well-defined inter-connected pore channels, as intended. The pore channels beneath the surface were not substantially blocked. The figure also demonstrates the effectiveness of how the zirconia slurry penetrated deep inside the sacrificial polymer template, proving that reticulated porous zirconia at 60 PPI can be prepared.

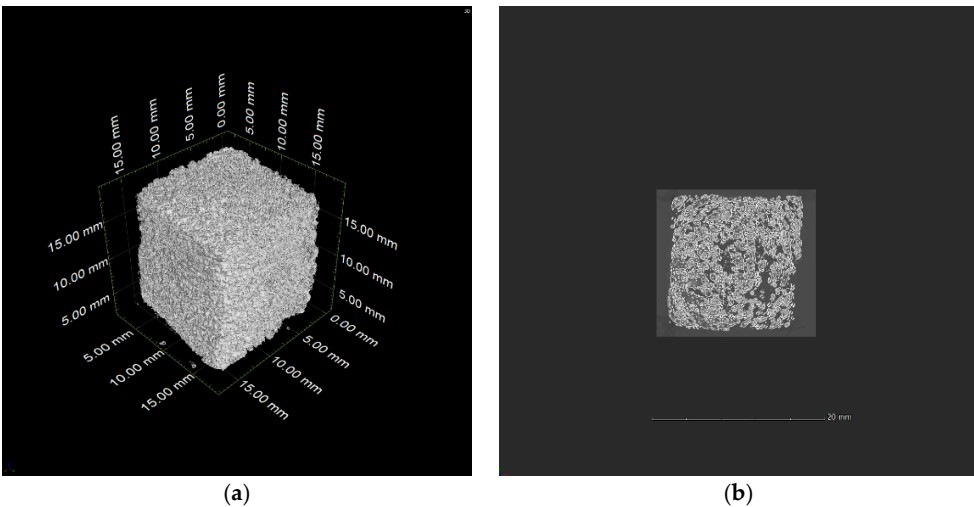

(a)                                    (b)

**Figure 4.** (**a**) Three-dimensional microstructure of a typical reticulated porous zirconia specimen (pore density = 60 PPI), reconstructed by μ-CT; and (**b**) representative slice extracted from the μ-CT reconstructions.

The particle size distributions of the zirconia, denoted here as as-received, coarse, intermediate, and fine, is shown in Figure 5. Zirconia particles with an appropriate particle size were selected for use here. If the particle size is too coarse, densification by sintering will be suppressed. Conversely, a nano-sized particle size is well densified, but agglomeration becomes severe such that sintering shrinkage and cracks are generated, reducing the compressive strength. Thus, zirconia with a sub-micron particle size was judged to be appropriate through other experimental results by the present researchers [26]. The particle size is a process variable for obtaining the optimum composition of reticulated porous zirconia specimens. As the particle size decreases, the particle size distribution decreases to 1.435 μm, 1.236 μm, 1.020 μm, and to 0.701 μm relative to the values above.

Figure 6a shows the XRD patterns of the phases of 3 mol% yttria-stabilized zirconia (3YSZ) with different sintering temperatures. In this study, yttria-stabilized zirconia (YSZ) with the addition of yttria was used. YSZ is well known to prevent the generation of cracks by a phase transformation (tetragonal phase to monoclinic phase) during cooling. For sintered 3YSZ at 1400, 1500, and 1600 °C, it was observed that the main phase is tetragonal (t), and no monoclinic (m) phase of $ZrO_2$ was observed. It was confirmed to have a stable tetragonal phase even at room temperature after cooling. The linear shrinkage outcomes as the sintering temperature is increased are shown in Figure 6b. As the sintering temperature

increases, the shrinkage rate increases. This can be judged as stemming from a decrease in the number of open pores due to densification between the particles as the sintering temperature increases. The density change of the reticulated porous zirconia with an increase the sintering temperature is shown in Figure 6c. Regardless of the change in the sintering temperature, it can be seen that the density increases as the PPI value of the sacrificial polymer template increases. The increase in the density with an increase the sintering temperature and PPI values can be determined to stem from the reduced pore size as the PPI values increase. Figure 6d shows the pore size distributions of the reticulated porous zirconia specimens (pore density = 45 PPI). The integral area is visualized with a quantity of mercury penetrating into the porous interior. At this point, mercury cannot penetrate into the closed pores; it can only penetrate into the open pores. As the sintering temperature increases to 1400 °C, 1500 °C, and 1600 °C, the mercury penetration decreases to 0.142 mL/g, 0.138 mL/g, and 0.097 mL/g, respectively. Because the number of strut walls is fixed, the difference in the pore size distribution varies with the sintering temperature. It is judged that the number of open pores decreased due to densification between the particles as the sintering temperature was increased.

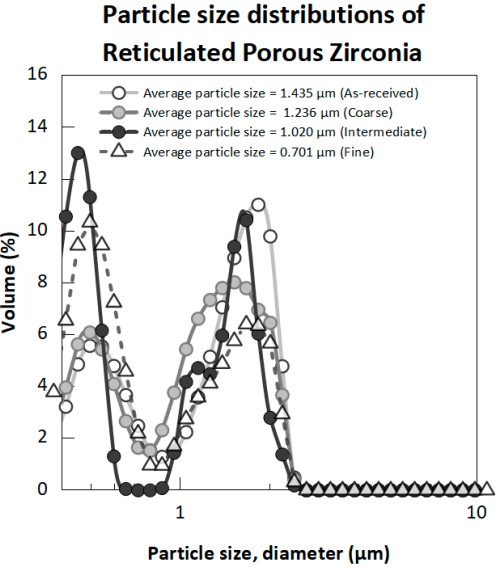

**Figure 5.** Particle size distributions of zirconia, for as-received, coarse, intermediate, and fine.

The microstructures of the reticulated porous zirconia when sintered at 1400 °C, 1500 °C, and 1600 °C are shown in Figure 7a–c, respectively. As the sintering temperature was increased, the average particle size increased. In addition, it can be seen that as the sintering temperature increases, the number of open pores decreases due to densification between the particles. When the reticulated porous zirconia is sintered at 1400 °C, pores exist and necks form between the particles. When the sintering temperature was 1500 °C, the grain boundaries between the particles became connected. When sintered at 1600 °C, the grain size grew because there were no pores that interrupted the grain growth. Figure 7d shows the microstructure of the reticulated porous zirconia when reticulated porous zirconia specimens were prepared from fine zirconia particles and sintered at 1600 °C. As the average particle size decreased, the particles agglomerated. The initial average particle size and particle distribution reportedly play very important roles during the sintering process of ceramics. As the average particle size becomes smaller, the surface energy increases, possibly leading to densification, but aggregation occurs due to electrostatic attraction.

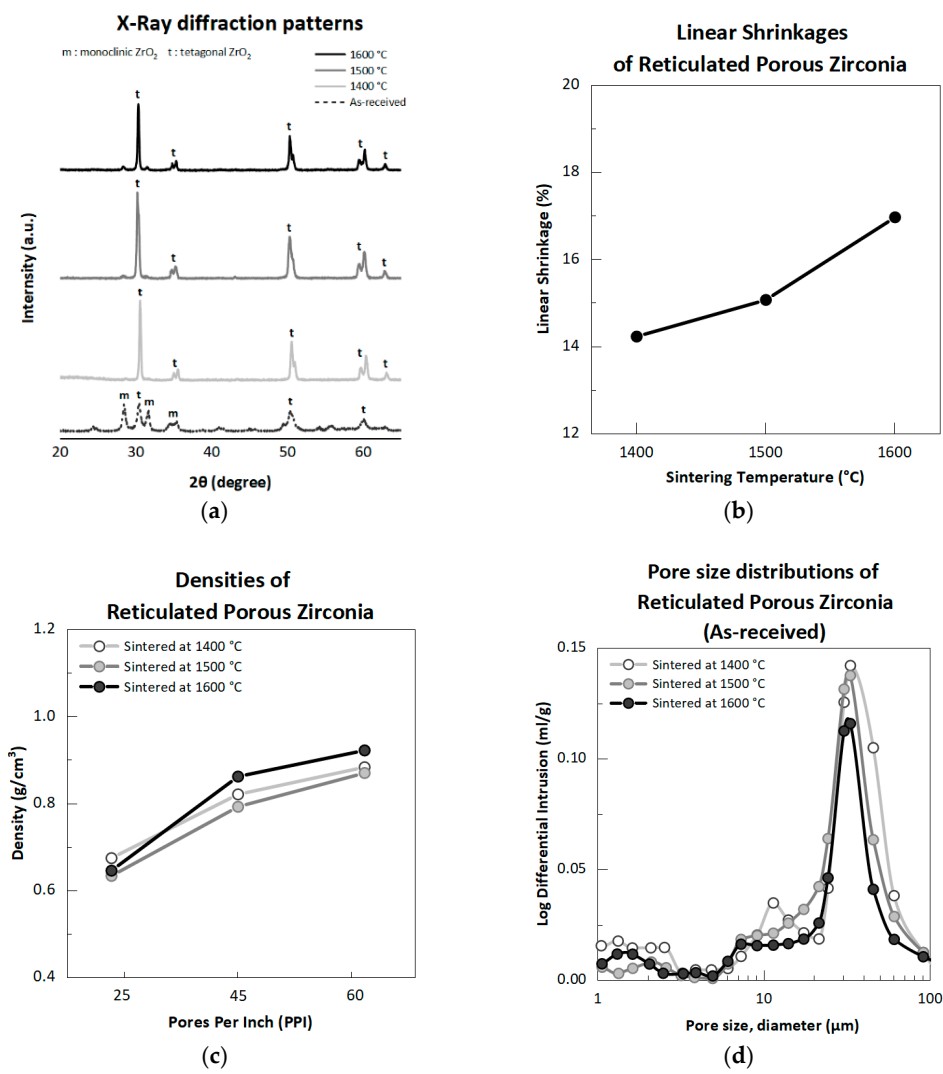

**Figure 6.** (**a**) X-ray diffraction patterns of the reticulated porous zirconia with different sintering temperatures, (**b**) linear shrinkage levels, (**c**) densities, and (**d**) pore size distributions of the reticulated porous zirconia samples when sintered at 1400, 1500, and 1600 °C.

Zirconia slurries were prepared with a combination of a dispersant (Dolapix CE 64), a thickener (MC), and an organic binder (PVA). The dispersant is vital to remove agglomerated zirconia particles and to prepare a uniform zirconia slurry. The organic binder increases the viscosity of the suspension and prevents the formation of cracks after the coating step while also maintaining the shape of the sacrificial polymer template after sintering. The thickener was used to increase the viscosity of the zirconia slurry and to coat the strut walls of the sacrificial polymer template tightly so as to minimize defects that can be caused by the strut walls. Without the addition of an organic binder and thickener, the viscosity of the zirconia slurry is low, making it difficult to coat the strut walls of the sacrificial polymer template tightly. In addition, in order to ensure that the viscosity can sufficiently coat the strut walls of the sacrificial polymer template tightly, zirconia particles must be added indefinitely until the desired viscosity is reached. However, doing this is not efficient, as noted above.

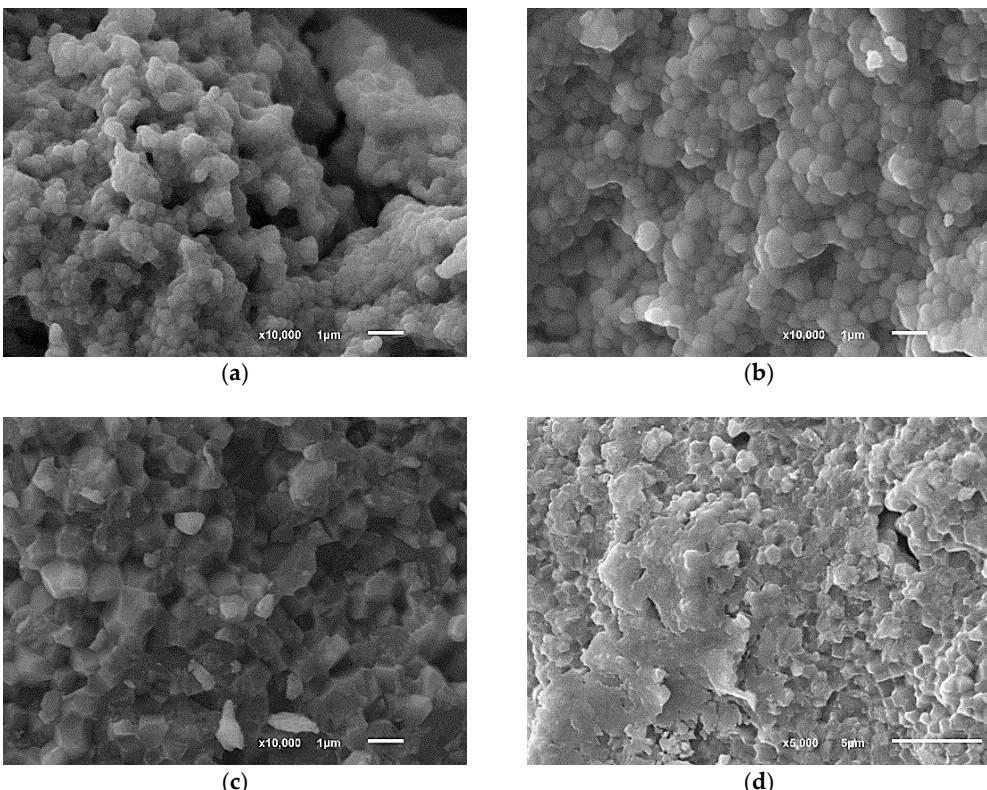

(a)　　　　　　　　　　　　　　(b)

(c)　　　　　　　　　　　　　　(d)

**Figure 7.** Typical SEM image of the reticulated porous zirconia samples, sintered at (**a**) 1400 °C, (**b**) 1500 °C, (**c**) 1600 °C, and (**d**) 1600 °C; these reticulated porous zirconia specimens were prepared from fine zirconia particles.

Figure 8a shows a plot of the viscosity vs. the shear rate of the zirconia slurries prepared with different average particle sizes. As the average particle size decreases due to ball-milling, the specific surface area increases and the points at which the PVA adheres to the particles increase. Therefore, as the average particle size increases, the viscosity increases. Overall, however, it is apparent that the viscosity tends to decrease as the average particle size decreases. Dolapix CE 64 has a lower molecular weight compared to other dispersants, is completely dissociated in water, and has an electrostatic stabilization effect immediately after it is added [27]. Therefore, it can be judged that the rheological properties of the slurry are optimized. To prepare reticulated porous zirconia specimens with a uniform structure using the replica template method when a sacrificial polymer template is impregnated into the zirconia slurries, proper viscosity and fluidity of the zirconia slurry are important. However, there are several challenges when preparing reticulated porous zirconia specimens under specific conditions. If the viscosity of the zirconia slurry is too high and the fluidity is reduced, the zirconia slurry cannot be completely coated onto the strut walls inside the sacrificial polymer template. Therefore, as shown in Figure 8b, voids arise inside the sacrificial polymer template, reducing the compressive strength. In contrast, if the viscosity of the zirconia slurry is too low and the fluidity is increased, the coating layer of the sacrificial polymer template strut walls becomes non-uniform and very thin. Therefore, the shape of the sacrificial polymer template cannot be maintained, making it difficult to complete the reticulated porous zirconia specimen. In addition, the low viscosity causes the number of cracks and defects to increase as the strut walls are completely uncoated. This reduces the compressive strength of the reticulated porous zirconia.

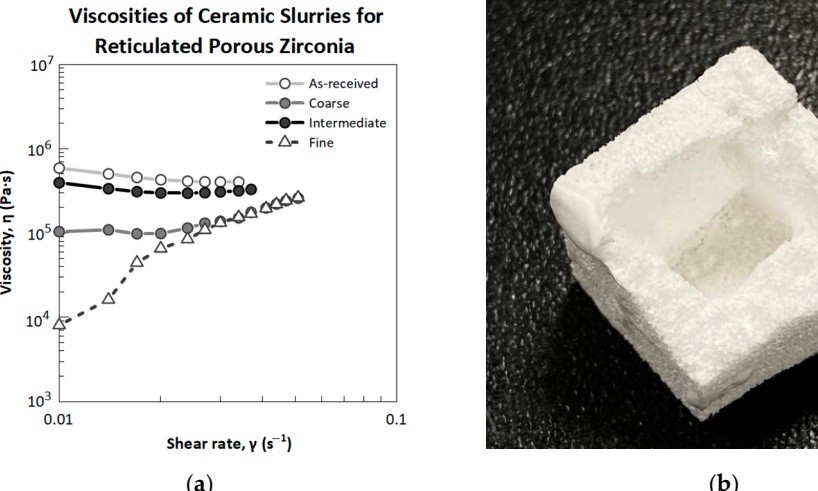

**Figure 8.** (**a**) Viscosities of the zirconia slurries for the preparation of reticulated porous zirconia: as-received, coarse, intermediate, and fine zirconia particles; (**b**) fractured reticulated porous zirconia specimen with a pore density of 80 PPI, which possesses internal macro-voids.

When the reticulated porous zirconia was sintered at 1400 °C, the compressive strength outcomes for each of the sample, from the as-received, coarse, intermediate, and fine zirconia particles, according to the change in the PPI value, are shown in Figure 9a. It has been reported that the larger the average particle size difference is, the more different the sintering behavior itself, and the finer the particles are, the more decreased the compressive strength becomes [26]. In addition, fine particles generally consist somewhat of agglomerated particles, causing non-uniformity and thus reducing the densification when sintered [28]. However, in this study, because the zirconia average particle size difference was not great, the compressive strength increased with the fine average particle size. In addition, as shown in Figure 4b, the compressive strength increases to 60 PPI because the zirconia slurries are uniformly coated onto the strut wall inside of the sacrificial polymer template. Figure 9b shows the density outcome when the reticulated porous zirconia is sintered at 1400 °C. Overall, this figure shows a trend identical to that of the compressive strength.

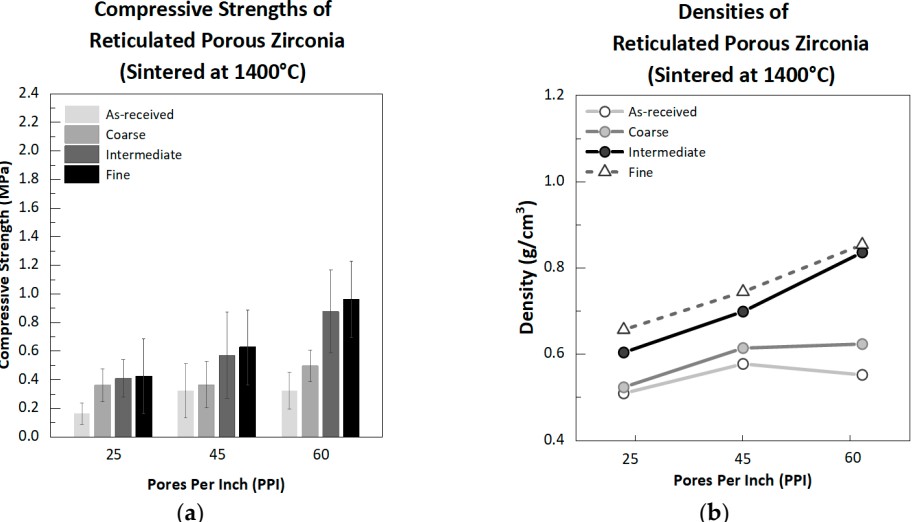

**Figure 9.** (**a**) Compressive strength outcomes of the reticulated porous zirconia prepared from the as-received, coarse, intermediate, and fine zirconia particles when sintered at 1400 °C, and (**b**) the corresponding density levels.

When the reticulated porous zirconia was sintered at 1500 °C, the compressive strength outcomes for each of the cases, from the as-received, coarse, intermediate, and fine zirconia particles, according to the change in the PPI value, are shown in Figure 10a. Overall, as the average particle size decreases, the compressive strength increases. Furthermore, as the degree of the coating of the strut walls with the zirconia slurries increases (as the PPI approaches 60), the compressive strength also increases. It can be seen that the zirconia slurries can be coated up to 60 PPI with the sacrificial polymer template. Figure 10b shows the densities when the reticulated porous zirconia is sintered at 1500 °C. As the PPI increases and the average particle size decreases, the density increases.

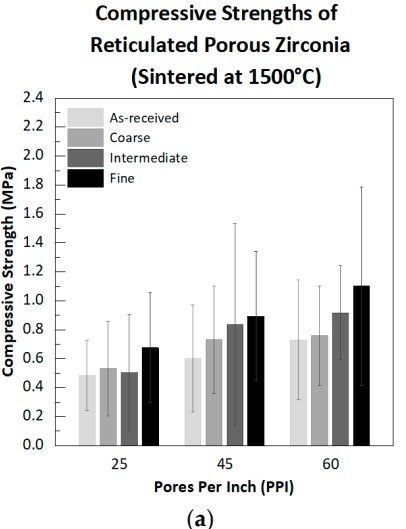
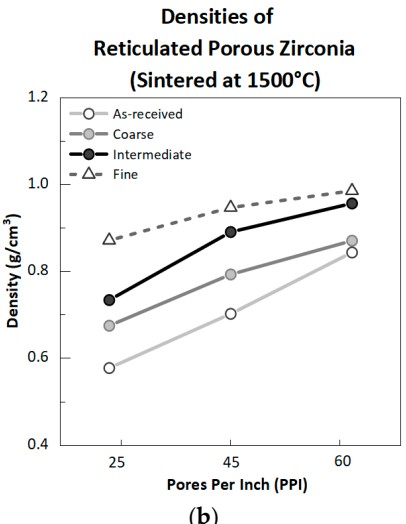

| (a) | (b) |

**Figure 10.** (**a**) Compressive strength outcomes of the reticulated porous zirconia prepared from the as-received, coarse, intermediate, and fine zirconia particles when sintered at 1500 °C, and (**b**) the corresponding density levels.

When the reticulated porous zirconia was sintered at 1600 °C, the compressive strength outcomes for each case, from the as-received, coarse, intermediate, and fine zirconia particles, according to the change in the PPI value, are shown in Figure 11a. As the average particle size decreases and the PPI increases, the compressive strength increases. The results of measurements of the compressive strength at 1400 °C, 1500 °C, and 1600 °C indicate that the compressive strength when the specimens are sintered at 1600 °C is the best. In this case, 80 PPI was utilized only at 1600 °C to obtain the maximum compressive strength. To maximize the compressive strength, samples of 80 PPI sintered at 1600 °C were measured, but at 80 PPI, a decrease in the compressive strength was noted. As the pore density of reticulated porous zirconia approaches 80 PPI, it is difficult to coat the strut walls inside the sacrificial polymer template completely and to remove the excess zirconia slurry. In addition, several blocked, interconnected pores form, and voids are caused inside the sacrificial polymer template, lowering the compressive strength. Figure 11b shows the density levels when the reticulated porous zirconia is sintered at 1600 °C. Overall, this trend is identical to that of the compressive strength.

The pore size distributions of the reticulated porous zirconia specimens (pore density = 45 PPI) prepared from the as-received, coarse, intermediate, and fine zirconia particles when sintered at 1400, 1500, and 1600 °C are correspondingly shown in Figure 12a–c. Because the number of strut walls is fixed, the difference in the pore size distribution varies with the average particle size. The pore size and the pore volume of the reticulated porous zirconia did not decrease significantly because the average particle size difference was not significant. However, the number of pores decreased; thus, the compressive strength outcomes were found to have increased.

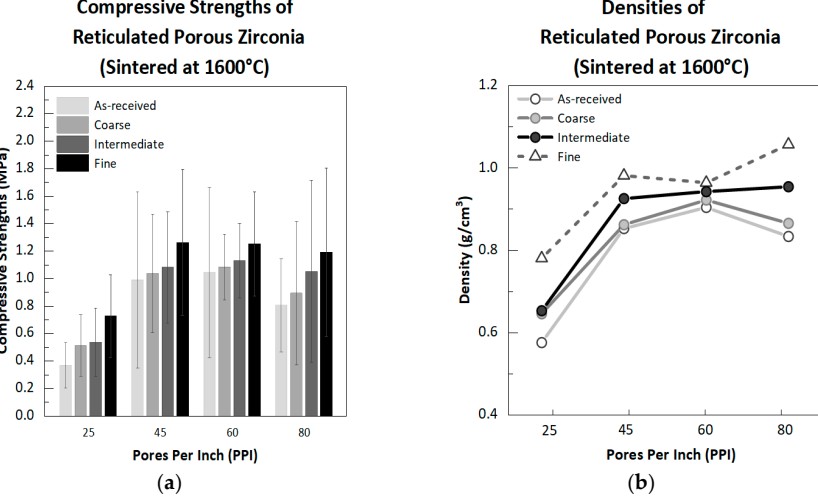

**Figure 11.** (**a**) Compressive strength outcomes of the reticulated porous zirconia prepared from the as-received, coarse, intermediate, and fine zirconia particles when sintered at 1600 °C, and (**b**) the corresponding density levels.

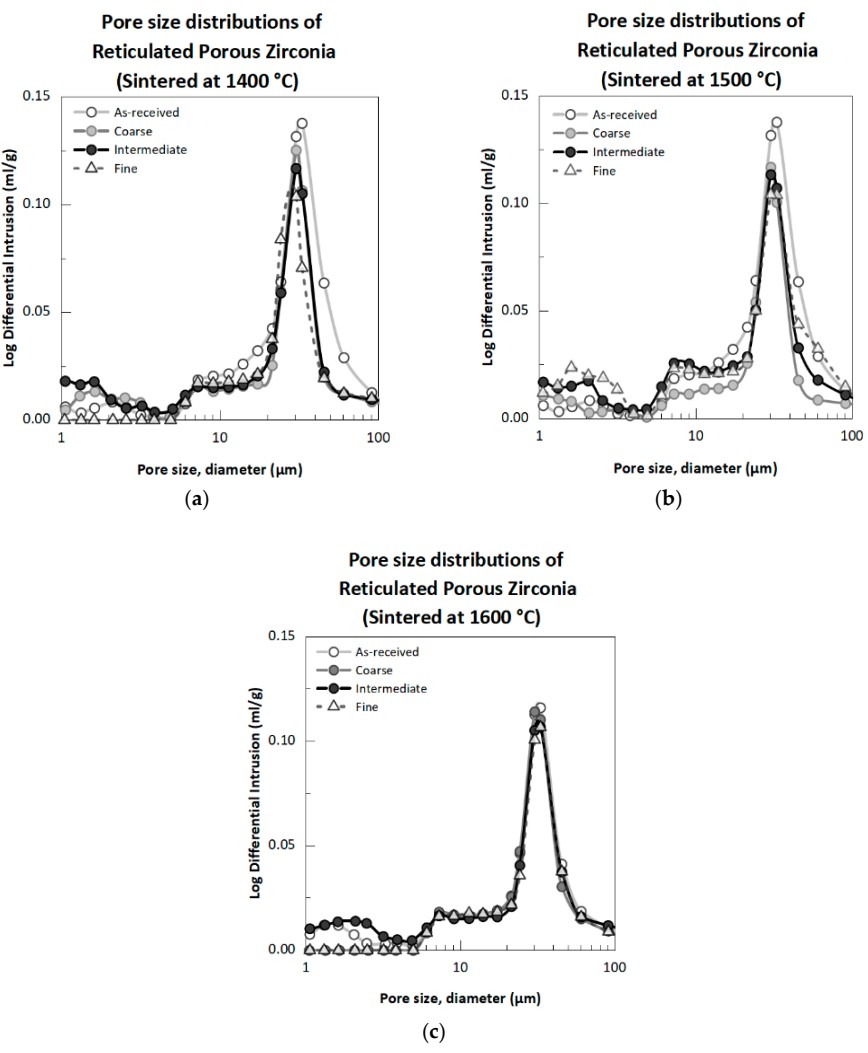

**Figure 12.** Pore size distributions of the reticulated porous zirconia prepared from the as-received, coarse, intermediate, and fine zirconia particles when sintered at (**a**) 1400 °C, (**b**) 1500 °C, and (**c**) 1600 °C.

Several conclusions can be drawn based on the results of the experiments conducted to obtain reticulated porous zirconia with high compressive strength outcomes. The main factors that can affect the compressive strength are the average particle size, the viscosity of the zirconia slurry, the sintering temperature, and the PPI of the sacrificial polymer template. First, it is possible to improve the compressive strength by controlling the average particle size of the zirconia. Fine particles generally consist somewhat of agglomerated particles and cause non-uniformity, thus reducing the degree of densification when sintering takes place. However, in this study, the compressive strength increased when the average particle size was fine because the average particle size was in the sub-micron range. Second, if the zirconia slurry viscosity is too high in the case of a sacrificial polymer template with a low PPI value (such as 25 PPI), the strut wall thickness increases, resulting in an increase in the compressive strength. However, at a high PPI value (such as 80 PPI), the zirconia slurry cannot be uniformly coated onto the sacrificial polymer template strut walls, resulting in macro-voids forming deep inside (as shown in Figure 8b). Conversely, if the viscosity of the zirconia slurry is too low, the likelihood of the generation of macro-voids is reduced but the thickness of the sacrificial polymer template strut walls becomes too thin and nonuniform, resulting in an expected decrease in the compressive strength. Therefore, when a uniform coating with the sacrificial polymer template with an optimized zirconia slurry is achieved, it is possible to improve the compressive strength. Third, it is possible to prepare reticulated porous zirconia with a wide range of strength values by controlling the sintering temperature. As the sintering temperature increases, the compressive strength increases due to densification between the particles. Finally, when all other conditions, such as the average particle size and sintering temperature, are fixed, the PPI of the sacrificial polymer template can be adjusted to increase the compressive strength. However, as shown in Figure 8b, sacrificial polymer templates with high PPI values, such as 80 PPI, have many strut walls and small pore sizes, resulting in interior macro-voids, thus reducing the compressive strength. To increase the compressive strength at 80 PPI, the slurry viscosity must be reduced, which is meaningless because the expected strength itself is lowered. In conclusion, it is possible to prepare reticulated porous zirconia having a strength outcome over a wide range (0.16~1.26 MPa) as needed by adjusting the main factors.

## 4. Conclusions

In this study, reticulated porous zirconia specimens were prepared using the replica template method. The reticulated porous zirconia specimens were prepared from as-received, coarse, intermediate, and fine zirconia particles. Different sintering temperatures (1400, 1500, and 1600 °C) and the effects of the average particle size on the viscosity of the zirconia slurry and the pore density of the sacrificial polymer templates (25, 45, 60, and 80 PPI) were assessed.

The optimum process conditions for each sintering temperature (1400, 1500, and 1600 °C) were determined. When the reticulated porous zirconia is sintered at 1400 °C with a pore density of 60 PPI and fine zirconia particles, the compressive strength was approximately 0.96 MPa. When the reticulated porous zirconia is sintered at 1500 °C with a pore density of 60 PPI and fine zirconia particles, the compressive strength reaches nearly 1.10 MPa. Finally, when the reticulated porous zirconia is sintered at 1600 °C with a pore density of 45 PPI and fine zirconia particles, the compressive strength becomes approximately 1.26 MPa. As the sintering temperature increases, the compressive strength increases due to densification between the particles.

Consequently, when the raw material was ball-milled, the compressive strength was improved compared to when it was not. In addition, at a higher sintering temperature, the sintering driving force increases, resulting in greater densification and thus increased compressive strength of the reticulated porous zirconia. These outcomes demonstrate that it is possible to prepare reticulated porous zirconia with a wide range of strengths

(0.16~1.26 MPa) as needed with an average particle size and while properly controlling the sintering temperature.

**Author Contributions:** J.-H.H. and K.-S.M. conceived and designed the experiments; C.-Y.L. and S.L. performed the experiments; J.L. and J.-H.H. analyzed the data; I.-H.S. contributed reagents/materials/ analysis tools; C.-Y.L. wrote the paper. All authors have read and agreed to the published version of the manuscript.

**Funding:** This research was funded by Fundamental Research Program of the Korean Institute of Materials Science (KIMS), Grant No. PNK7420, and by National R&D Program through the National Research Foundation of Korea (NRF) funded by Ministry of Science and ICT (2020M3H4A3106359).

**Institutional Review Board Statement:** Not applicable.

**Informed Consent Statement:** Not applicable.

**Acknowledgments:** This research was funded by Fundamental Research Program of the Korean Institute of Materials Science (KIMS), Grant No. PNK7420 and by National R&D Program through the National Research Foundation of Korea (NRF) funded by Ministry of Science and ICT (2020M3H4A3106359).

**Conflicts of Interest:** The authors declare no conflict of interest.

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
