# Peer review of "Effect of the Sintering Temperature on the Compressive Strengths of Reticulated Porous Zirconia"

_applsci, doi:10.3390/app11125672_

Round 1

Reviewer 1 Report

  1. Review for

    applsci-1264088

    Effect of the Sintering Temperature on the Compressive Strengths of Reticulated Porous Zirconia

  2. The study of reticulated porous zirconia is important. Especially for the compressive strength. According to the description in this article, the authors concluded a good approach to improve the compressive strength by evaluating different processing methods. I think, nonetheless, that the manuscript could be improved if the authors could address the comments and recommendations I listed below.
  3. Brief state your research conclusion in the Abstract.
  4. Line 101: Some details about your SEM set-ups may need. Such as Voltage, Working distance, and Image Mode(SE or BSE)
  5. Present your micro-CT images (Figure 4) in an aesthetic & professional way. Especially for Figure 4b which looks too casual.
  6. Your experiment phenomena are well described in the content and your results are very informative in practical application. However, the lack of in-depth mechanism analysis makes your article scientifically poor. 
  7. My decision is to recommend for publication with appropriate editing.

Reviewer 2 Report

The research work on the effect of sintering temperature on the compressive strength of mesh porous zirconium oxide is very interesting. The paper is well constructed and does merit publication.  The discussion of the results was properly described although in some places the discussion should be a bit more extensive.

However, I have a few comments as follows:

1) In my opinion, the authors should include  X-ray Diffraction technique is used to obtain microstructural information of the samples with the spectrum analysis.

2) Why the authors did not use the FE-SEM method for imaging the sample surface ? This technique also allows the obtaining of microstructural informationl, in quantitative and qualitative composition analysis.

3) Why did the authors use yttrium to stabilize zirconia?

Therefore, I recommend publication of this manuscript after some minor revision that could additionally improve this paper.
